# Penalizing the High-likelihood: A Novel Sampling Method for Open-ended Neural Text Generation via Inverse Probability Weighting

## Abstract

Traditional stochastic sampling methods for open-ended neural text generation focus on truncating the low-likelihood part of the predicted distribution. They do not directly manipulate the high-likelihood part, which leads to the likelihood trap that induces repetition and boredom. They also do not directly leverage that human does not always favor high-likelihood texts. Inspired by these, we propose a novel sampling method that rescales the high-likelihood part of the distribution with inverse probability weighting. It increases the diversity by rescaling and penalizing the high-likelihood words, and preserves the fluency by using multi-filtering truncation on the low-likelihood words. We use pre-trained language models to compare our algorithm with traditional sampling methods. Results show that our algorithm can significantly increase the diversity and novelty of generated texts without corrupting the fluency.

## 1 Introduction

Open-ended neural text generation is greatly affected by decoding methods. Counter-intuitively, the quality-oriented decoding methods such as beam search, which maximizes the likelihood of decoded texts, induces the well-known *text degeneration* (Holtzman et al., 2020; Welleck et al., 2020) and *likelihood trap* (Zhang et al., 2021; Basu et al., 2021), that is, the high-likelihood texts are prone to be repetitive and boring with low quality. As a result, many works have focused on stochastic sampling method such as top-$k$ sampling (Fan et al., 2018; Holtzman et al., 2018) or nucleus sampling (top-$p$ sampling, Holtzman et al., 2020). These methods first truncate the low-likelihood part of the language model's predicted distribution, then perform stochastic sampling on the truncated distribution for all decoding time steps. Other methods, such as temperature sampling, rescale the log-likelihood of *all* words to control the quality of generated texts. Recent works (Caccia et al., 2020; Nadeem et al., 2020; Zhang et al., 2021) reveal that these methods achieve on-par performance regarding their quality-diversity trade-off feature. Still, there exist undiscovered properties to understand better the relationship between stochastic sampling algorithms and open-ended neural text generation (Nadeem et al., 2020).

We note that none of the traditional sampling algorithms have directly manipulated the high-likelihood part of the distribution since high-likelihood words are always considered to be "trustworthy". Essentially, the observed quality-likelihood curve by human judgment is inversely proportional to the likelihood in the high-likelihood area (Zhang et al., 2021), which confirms the intuition that human does *not* always favor high-likelihood words (Holtzman et al., 2020; Welleck et al., 2020). Inspired by these, we propose a novel sampling method, namely the *interquartile range inverse probability* (IQR-IP) sampling algorithm. It increases the diversity of generated texts by rescaling and penalizing the high-likelihood part of the predicted distribution with inverse probability weighting and preserves the fluency by using multi-filtering truncation on the low-likelihood. The rescaled distribution will achieve a closer resemblance to the quality-likelihood curve (such as the human judgment of Figure 1 by Zhang et al., 2021), as is illustrated in Figure 1. Empirical results show that our algorithm can increase the diversity and novelty of generated text without corrupting the fluency.

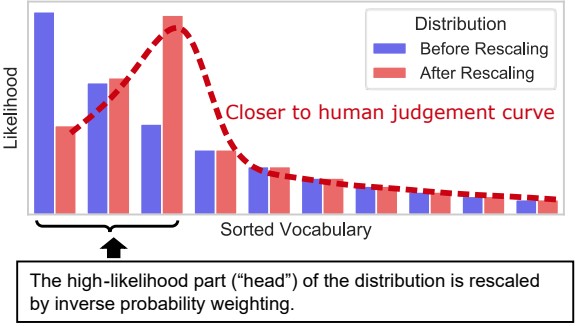

Figure 1: Illustration of our algorithm. The high-likelihood part of the language model's predicted distribution on each sampling step is rescaled by inverse probability weighting to penalize the high-likelihood words. The rescaled distribution (colored in red) will achieve a closer resemblance to the quality-likelihood curve (see the human judgment curve of Figure 1 by Zhang et al., 2021).

## 2 THE LIKELIHOOD TRAP

### 2.1 TRAPPED TRAJECTORY INDUCED BY THE HIGH-LIKELIHOOD

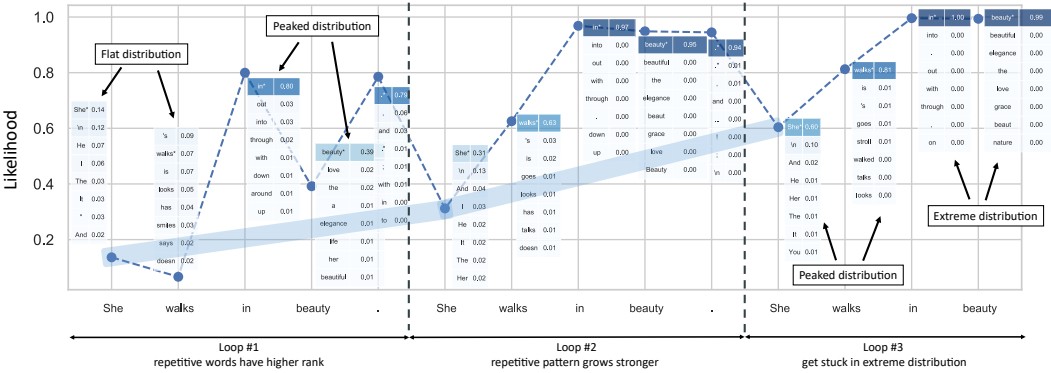

Figure 2: The trajectory of predicted probability ("o" marker) and predicted distribution (heatmap box beside each marker in "word-likelihood" format, with the sampled word marked by "*") for the first three repetition loops. It contains infinite repetitive loops of "She walks in beauty." (with a generated period). The trajectory of the repetitive word "She" is highlighted in shadow, which shows the increase of predicted probability and the gradually peaked predicted distribution.

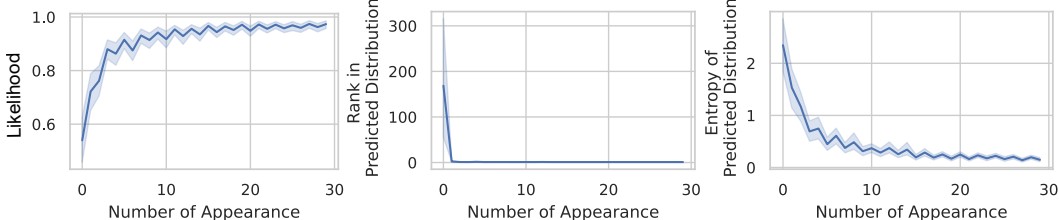

Figure 3: Trajectories of repetitive words extracted from samples that contain repetition loops. Repetitive words that appear more than 30 times are extracted and aligned to form their trajectories. A few appearances of repetitive words quickly lead the model to extreme distribution that causes repetition loops.

We first study the likelihood trap in open-ended generation cases. Unlike existing works, we are curious about the generation trajectory continued from a sharing context. So we use GPT-2 Small

Radford et al. (2019) with nucleus sampling ($p = 0.95$) to generate 5,000 samples using the same prompt. We choose the prompt "She walks in beauty" (from Lord Byron's poetry) to set a high-novelty reference. To detect trapped repetitions on the generated passages, we adopt the *n*-gram entropy metric (Shannon & Weaver, 1963; Zhang et al., 2018; He & Glass, 2020) by calculating the entropy of *n*-gram distribution in a fixed-length token window. Empirically, we found that the entropy threshold of 2.0 for unigram on 200-length token windows is good enough to filter repetition. We present a generated passage that contains *infinite loops* of the prompt, and the generation process gets trapped in repeating the input prompt. The likelihood trajectory of first 3 loops is presented in Figure 2. We report the following observations.

- Repetitive words always have *high likelihood* and *high rank* in the predicted distribution (see "*" labeled words in each heatmap box in Figure 2).

- Repetition tendency *grows stronger* when *more loops occur* (due to a few sampling steps that happen to pick repetitive token in non-extreme distribution, e.g, in Loop #2), as the *flat distribution* in Loop #1 (e.g., "She" and "walks") gradually becomes *peaked distribution* in Loop #3, and peaked distribution in Loop #1 (e.g., "in" and "beauty") becomes *extreme distribution* in Loop #3, which reciprocally contributes to stronger repetition pattern in the context.

- The predicted distribution got *stuck* in *extreme distribution* that assigns almost all probability mass for repetitive words (e.g., "in" and "beauty" in Loop #3).

To further verify these phenomena, we extract and align the trajectories of each repetitive word that occurs more than 30 times in the context from all generated passages to observe its overall trajectory. Figure 3 presents the trajectories of likelihood, rank in predicted distribution, and entropy of predicted distribution, where $x$ axis is the number of the appearance of repetitive words. After a few appearances of repetitive words, the predicted distribution will quickly get stuck in extreme distribution where predicted probability approaches 1, rank approaches 1, and entropy approaches 0, rendering infinite repetition loops. The undesired behavior of high-likelihood words on the predicted distribution induces the likelihood trap and leads the model to exhibit repetition behavior.

## 2.2 Improving Diversity by Penalizing the High-likelihood

We present a detailed observation of the high-likelihood words in Figure 4. It shows that lower-likelihood words on a flat distribution are reasonable choices. If we rescale the distribution and emphasize these lower-likelihood words to improve the diversity and novelty, the fluency of generated passage will not be compromised. Besides, it is proven beneficial to increase generation diversity by emphasizing less probable words during training (Welleck et al., 2020). Furthermore, human judgment exhibits an inverse correlation to the likelihood in the high-likelihood part (Figure 1, Zhang et al., 2021). Inspired by these, we adopt the *inverse probability weighting* method that is commonly seen in causal inference (see Chapter 2, Hernán MA, Robins JM, 2020). We first identify a small subset of high-likelihood words that contains all reasonable choices (such as in Figure 4). Then adopt inverse probability

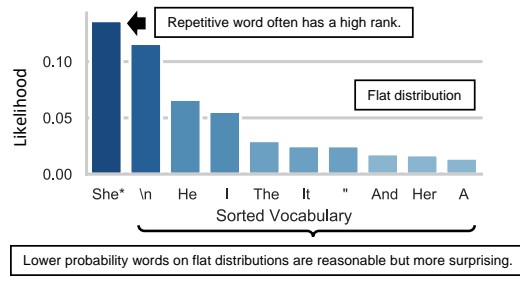

Figure 4: Illustration of the high-likelihood "head" on the flat distribution of the first sampling step of Loop #1 from Figure 2. Besides "She" that has the highest predicted probability, lower probability words ("\n", "He", "I", "The", ...) are also reasonable.

weighting to rescale the distribution of the "head" and penalize the high-likelihood, as is illustrated in Figure 1.

# 3 INTERQUARTILE RANGE INVERSE PROBABILITY SAMPLING ALGORITHM

## 3.1 FINE-GRAINED FILTERING ON THE LOW-LIKELIHOOD

The primary difficulty in identifying the high-likelihood "head" to rescale is the variation of the shape of the predicted distribution, i.e., the discrepancy between the flat distribution and the peaked distribution (Holtzman et al., 2020). Intuitively, the *interquartile range* (IQR) can adapt to such variation since it is based on quantile. Furthermore, we also need to leverage the traditional filtering methods, which truncate low-likelihood words to preserve fluency and ensure that reliable words are kept to calculate IQR. As a result, we propose to perform fine-grained filtering on the low likelihood.

Let $p_{LM}(x_t|x_{1:t-1})$ denote the auto-regressive language model's predicted probability of word $x_t$ from vocabulary $V$ given its context $x_{1:t-1}$ on time step $t$ (Bengio et al., 2003). All the following manipulations are conducted across all possible $t$. For simplicity, we directly use $p(x)$ to represent $p_{LM}(x_t|x_{1:t-1})$. We propose to jointly filter an initial subset $V_{fil}$ out of $V$ using top-$k$ filtering (with parameter $k$) and nucleus filtering (with parameter $p$).

$$V_{fil} = \text{top-}k(V) \cap \text{nucleus-}p(V). \tag{1}$$

Let $p_{fil}(x)$ denote the normalized distribution on $V_{fil}$. We propose to calculate IQR of $p_{fil}(x)$, that is, calculate 75% percentile of $p_{fil}(x)$ as $Q_3$, 25% percentile as $Q_1$, let $IQR = Q_3 - Q_1$ (all scalar), and divide $V_{fil}$ into subsets by using likelihood threshold determined by IQR as follows.

**IQR Subset Division of $V_{fil}$:**

$$
\begin{aligned}
V^{VeryHigh} &: p_{fil}(x) \geq Q_3 + \rho \times IQR \\
V^{High} &: Q_3 + \rho \times IQR > p_{fil}(x) \geq Q_3 \\
V^{Medium} &: Q_3 > p_{fil}(x) \geq Q_1 \\
V^{Low} &: Q_1 > p_{fil}(x) \geq Q_1 - \rho \times IQR
\end{aligned}
\tag{2}
$$

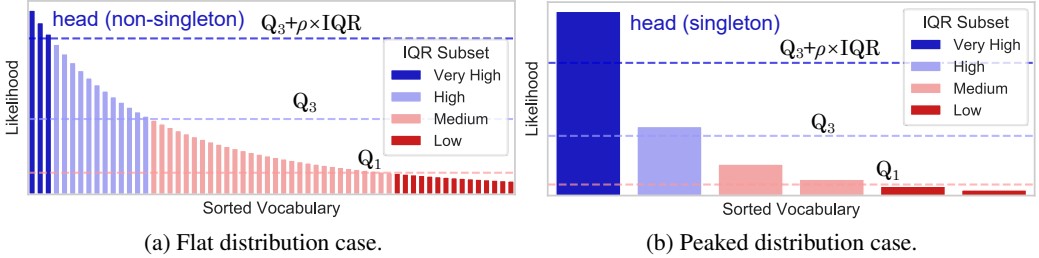

(a) Flat distribution case.    (b) Peaked distribution case.

Figure 5: Illustration of IQR subset division.

where $\rho$ is the hyperparameter for IQR coefficient with the typical value being 1.5. The division is illustrated in Figure 5. Considering the outlier-identifying nature of IQR, $V^{VeryHigh}$ can be regarded as the "head" part that we need to rescale, which we expect that the likelihood of the least probable word in $V^{VeryHigh}$ is still "high enough" to be reasonable choices (Figure 5a). Since IQR is based on the quantile, $V^{VeryHigh}$ will be singleton on peaked distribution that contains "unquestionably right" words (Figure 5b). In that case, manipulating and redistributing the probability mass of $V^{VeryHigh}$ does not have any effect. It will not corrupt peaked distribution cases with "unquestionably right" words.

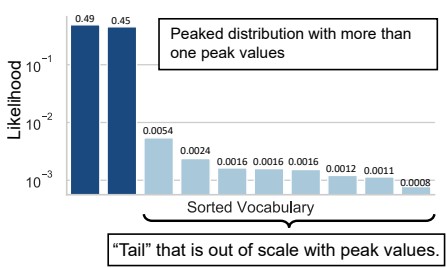

Figure 6: Peaked distribution with two peaks.

We also consider a particular case of distribution. Figure 6 presents an example of peaked distribution with more than one peak value. A small value of $p$ for nucleus sampling will miss the second peak, while a large value of $p$ will let in low-likelihood words that are out of scale with peak values. We note that it can be resolved by considering the scale constraint of likelihood. Concretely, we propose a novel filtering method by defining a scale threshold as the fraction of the maximum likelihood of the predicted distribution. We name it as the "Top-1 Controlled" (Top1CTRL) filtering with parameter $n$ as follows.

$$V^n = \{x \mid p(x) \geq \max p(x)/n, x \in V\}. \tag{3}$$

Note that a small value of $n$ might over-prune the vocabulary and harm the diversity. As a result, we propose to use $V^n$ to prune $V_{fil}$ in a fine-grained manner, as is described in Equation 4 and Figure 7. Case 1 ensures that $V^n$ does not over-prune words categorized as "Very High" or "High" since they are identified by IQR and are likely to contain reasonable words. Case 2 describes other cases where $V^n$ directly truncates $V_{fil}$ and works jointly with nucleus filtering and top-$k$ filtering. The pruned set is denoted by $V'_{fil}$. Empirically, $n$ can be set to a fixed value of $100$ to achieve good performance.

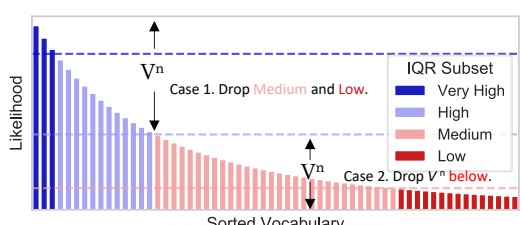

Figure 7: Illustration of Top1CTRL filtering.

$$V'_{fil} = \begin{cases} V^{VeryHigh} \cup V^{High}, & if \quad V^n \subseteq \left(V^{VeryHigh} \cup V^{High}\right) \quad \text{(Case 1)} \\ V_{fil} \cap V^n, & otherwise \quad \text{(Case 2)} \end{cases}. \tag{4}$$

### 3.2 Inverse Probability Weighting on the High-likelihood

With $V'_{fil}$ acquired, we propose to redistribute the probability mass for each word in $V^{VeryHigh}$ (i.e., the "head") proportionally to its inverse probability while keeping the sum of probability mass in $V^{VeryHigh}$ constant. Let $p'_{fil}(x)$ denote the normalized distribution on $V'_{fil}$. The transformation on $V^{VeryHigh}$ is described in Equation 5 and Figure 1, where $p_{inv}(x)$ denotes the rescaled distribution.

$$p_{inv}(x) = \begin{cases} \left(\sum_{x \in V^{VeryHigh}} p'_{fil}(x)\right) \times \dfrac{p'_{fil}(x)^{-1}}{\sum_{x \in V^{VeryHigh}} p'_{fil}(x)^{-1}}, & \forall x \in V^{VeryHigh} \\ p'_{fil}(x), & otherwise \end{cases}. \tag{5}$$

Finally, the sampling is performed with $p_{inv}(x)$. We refer to the above algorithm as the *interquartile range inverse probability* (IQR-IP) sampling algorithm. The main features of our algorithm are as follows.

**A.** We use fine-grained truncation on low-likelihood "tail" with 3 parameters ($p$, $k$, and $n$). It aims to control the "tails" to preserve fluency and guarantee the correct identification of the "head". Empirically, these parameters can be fixed around the reference point to achieve good performance.

**B.** The distribution of the high-likelihood "head" identified by IQR is rescaled by inverse probability weighting using Equation 5. It aims to improve diversity by penalizing the high-likelihood words, resembling the quality-likelihood curve of human judgment.

## 4 Empirical Results

To provide generalizable results, We use the pre-trained GPT-2 XL model released by Wolf et al. (2019) (without any fine-tuning) for text generation and evaluation. We set the generation length to be

200 tokens and generate 5,000 passages for each hyperparameter configuration using the same prompt in Section 2.1. We choose the commonly used nucleus sampling, top-$k$ sampling, and temperature sampling as baseline methods. The following automatic metrics are considered.

**Fluency**. We calculate the averaged perplexity (PPL) of the generated passages (Ippolito et al., 2019; Holtzman et al., 2020; Basu et al., 2021) to reflect fluency. Note that the metric does not equal quality since low-perplexity passages might be repetitive and boring, while high-perplexity passages might be unreasonable. Like most existing works, we compare the metric w.r.t the human-level metric.

**Diversity**. We first calculate the Self-BLEU (4 and 5) score (Zhu et al., 2018) that reflects the overlapping between different generated samples. We then calculate $n$-gram entropy (Zhang et al., 2018) that reflects the diversity of $n$-gram distribution and repetition tendency. We also calculate the Zipf coefficient (Zipf, 1949; Newman, 2005), a linguistic feature that reflects the sloping tendency of word frequency distribution on a corpus.

### 4.1 METRIC VARIATION WITH HYPERPARAMETERS

We first present the results of metric variation by tuning hyperparameters. As is shown in Figure 8, our algorithm achieves human-level PPL with *more strictly filtered vocabulary*, which means our algorithm truncates more low-likelihood "tails" and still achieves equal fluency to human text. It is a desirable feature since the "tails" that contain unreasonable words will lower the quality of the generated text.

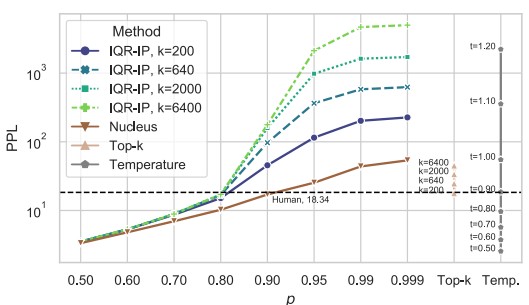

Figure 8: Results for the perplexity (PPL) of generated texts. They show that our algorithm achieves human-level fluency with less "tail" than traditional sampling algorithms. The horizontal line refers to human-level perplexity reported by Radford et al. (2019).

As is shown in Figure 9, the Self-BLEU scores achieved by our algorithm decrease significantly faster than traditional methods, which indicates great diversity gain. Note that it can achieve almost the same score with "pure sampling" (near nucleus sampling with $p = 0.999$, temperature sampling with $t = 1.0$), representing the upper bound of diversity for traditional methods. It suggests that the diversity boundary of traditional methods is limited, while our method effectively expands the diversity boundary. Similarly, results for 3-gram entropy in Figure 10a show that the entropy metric of our algorithm grows faster and achieves the human-level metric with less "tails". These results reveal that our algorithm achieves human-level diversity metrics by truncating more "tails" than traditional methods and compensating the diversity loss by rescaling the high-likelihood.

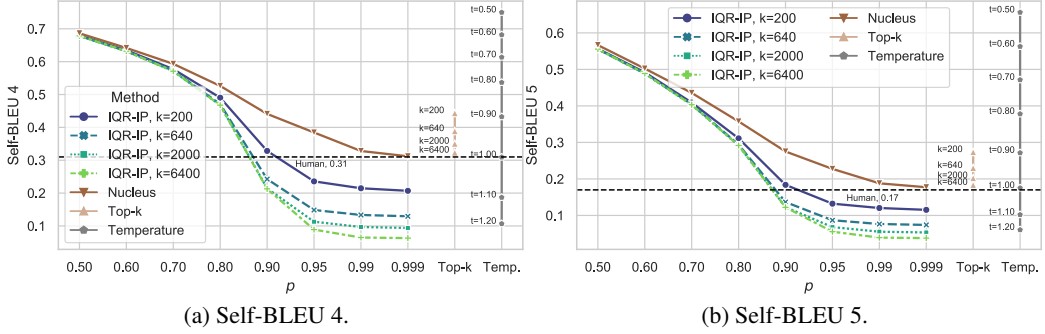

(a) Self-BLEU 4.                    (b) Self-BLEU 5.

Figure 9: Results for self-BLEU 4 and 5. They also show that our algorithm achieves human-level diversity with less "tail". The horizontal line refers to human-level self-BLEU scores reported by Holtzman et al. (2020).

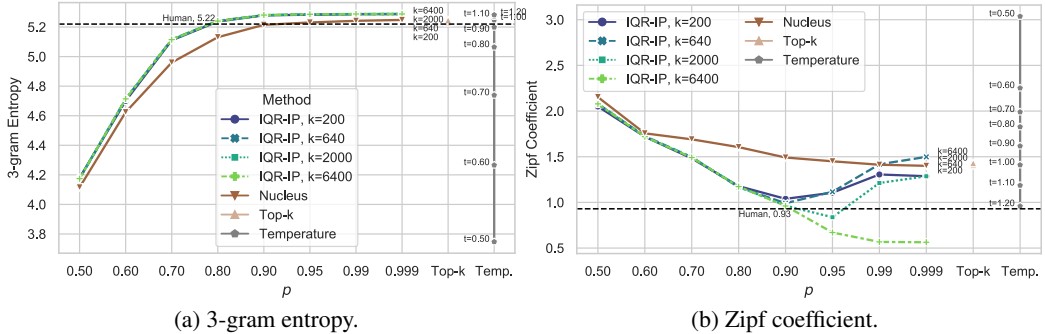

(a) 3-gram entropy.

(b) Zipf coefficient.

Figure 10: Results for 3-gram entropy and Zipf coefficient. Figure 10a also shows that our algorithm achieves human-level repetition entropy with less "tail". Horizontal line (5.22) refers to the metric of human text on the training dataset of WikiText-103 (Merity et al., 2017). Figure 10b shows that our algorithm can achieve the human-level Zipf coefficient while traditional sampling algorithms can't. The horizontal line refers to the human-level Zipf coefficient reported by (Holtzman et al., 2020).

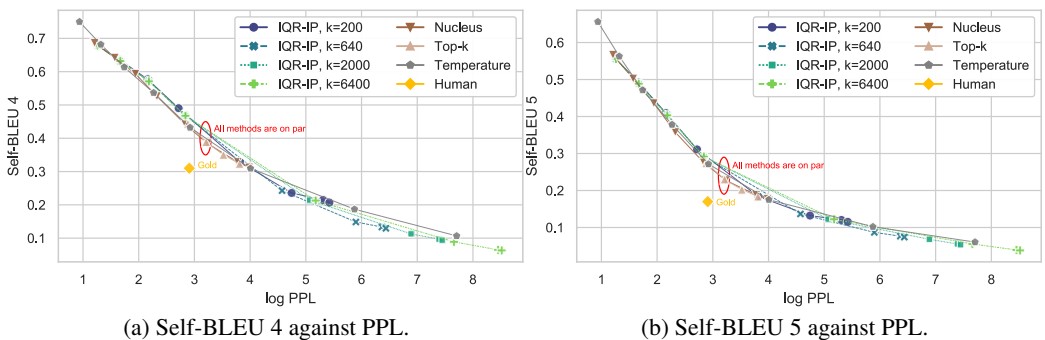

(a) Self-BLEU 4 against PPL.

(b) Self-BLEU 5 against PPL.

Figure 11: Trade-off curve of self-BLEU against PPL. They show that all methods are on par regarding the self-BLEU metric.

Results for the Zipf coefficient in Figure 10b are intriguing. They show that our algorithm can fit identical Zipf coefficient to human-level metric, while traditional sampling methods can't. It indicates that the rescaling transformation of our algorithm renders flatter and less concentrated distribution of words, which is closer to the human-level metric and unable to achieve by traditional sampling methods.

## 4.2 METRIC TRADE-OFF

Many existing works have investigated the metric trade-off curve to evaluate sampling algorithms. For example, Nadeem et al. (2020) state that violating the *entropy reduction property* or *slope preservation property* will result in drastic performance degradation on the quality-diversity plane (see Figure 3, Nadeem et al., 2020). Since our method violates them, we investigate the metric trade-off behavior by aligning each diversity metric (self-BLEU, Zipf coefficient, and 3-gram entropy) against the fluency metric (PPL) on the 2D plane. Results are shown in Figure 11 and Figure 12. Clearly, although our algorithm *does* violate all three properties by Nadeem et al. (2020), it still achieves on-par trade-off performance to traditional methods regarding self-BLEU and entropy. More importantly, Figure 12b shows that the Zipf coefficient trade-off curve of our method is considerably closer to the human-level point than all baseline methods, which can be inferred from the previous metric variation results. We boldly hypothesize that the properties by Nadeem et al. (2020) might not be necessary to design novel sampling methods. Instead, they might be boundaries to break for higher novelty, as our method rescales the high-likelihood "head" but achieves on-par or even better performance than baseline methods.

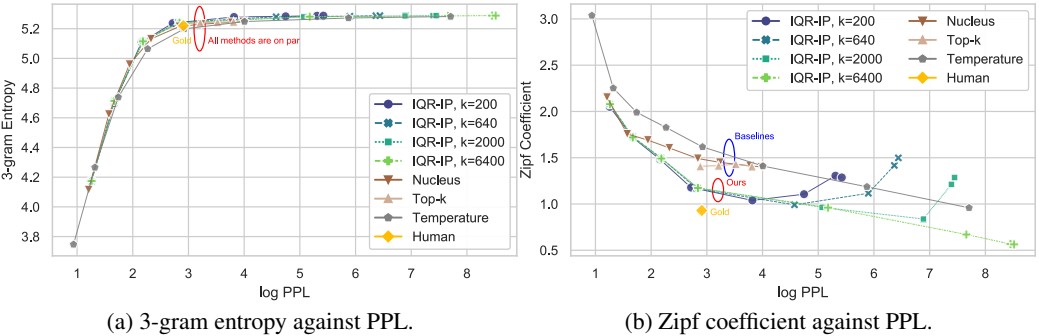

(a) 3-gram entropy against PPL.

(b) Zipf coefficient against PPL.

Figure 12: Trade-off curve of 3-gram entropy / Zipf coefficient against PPL. While all methods are on par regarding the entropy metric, the Zipf coefficient curve of our method is considerably closer to the human-level point than all baseline methods.

### 4.3 HUMAN EVALUATION

It is noteworthy that the quality of generated texts can be highly variable regarding the hyperparameter space as well as the stochastic nature of the sampling process. With a fixed sampling parameter, the generated texts yield a distribution of PPL with variational quality, as is shown in Figure 13. Such variation demands a large number of samples that sufficiently cover and represent the distribution, which requires an extremely high monetary cost to achieve meaningful results. Instead, we adopt an on-equal-footing fluency evaluation paradigm similar to Zhang et al. (2021). We set five pre-defined targets of PPL and filter the generated passages near these targets from all hyperparameter configurations for all sampling algorithms in a post-decoding manner. In that case, we can collect an equal number of filtered passages per PPL level per method, measured to be similarly fluent. We argue that it helps to cancel the quality variation issue for human evaluation. We also decompose the quality (overall) metric into the fluency metric and the novelty metric for human evaluation since high-likelihood passages with low quality are expected to be still fluent but boring. See Appendix A for details.

Following commonly paradigm (Ippolito et al., 2019; Nadeem et al., 2020; Zhang et al., 2021), we use Amazon Mechanical Turk for human evaluation. Results are shown in Table 1, which indicates that our algorithm achieves both higher fluency and novelty than traditional methods. They also revisit the conclusion that the three baselines achieve on-par performance. We present generated samples with PPL near the reference text in Table 2. Under the same PPL level, traditional methods favor creating comparatively *plain and narrative* passages, while our algorithm favors

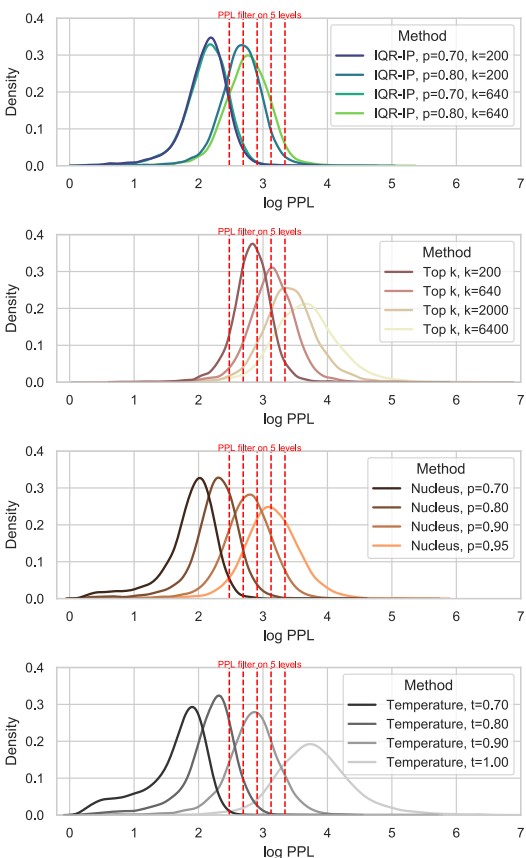

Figure 13: The estimated probability density function of PPL's distribution for selected parameter configurations. To cancel the impact of quality variation, we propose to use pre-defined PPL filters (colored in red vertical lines) to collect generated passages with PPL around these filters (PPL $\pm 0.5$ on each level) in a post-decoding manner.

creating *novel and surprising* passages. We relate these results to automatic results of diversity metrics by aggregating the filtered passages from all PPL levels per sampling algorithm to report their overall diversity metrics in Table 1. They show that under the on-equal-footing fluency paradigm, all methods are on par with each other regarding self-BLEU and 3-gram entropy. However, our method achieves a significantly lower Zipf coefficient, which confirms previous results. It reveals the nature of our method. Compared to traditional methods, our method dramatically flattens the word distribution of the generated passages (with a lower Zipf coefficient) by penalizing the high-likelihood words, which achieve similar fluency but exhibit higher diversity and novelty.

| Method | SB-4 ↓ | SB-5 ↓ | ZC ↓ | Ent-3 ↑ | Fluency ↑ | Novelty ↑ | Overall ↑ |
|---|---|---|---|---|---|---|---|
| Nucleus | 0.31 | 0.18 | 1.49 | 5.24 | 3.70 | 3.59 | 3.65 |
| Top-*k* | **0.30** | **0.17** | 1.39 | 5.24 | 3.79 | 3.49 | 3.64 |
| Temperature | 0.32 | 0.19 | 1.60 | 5.23 | 3.74 | 3.53 | 3.64 |
| IQR-IP (*ours*) | 0.31 | 0.18 | **1.19** | **5.25** | **3.83** | **3.67** | **3.75** |

Table 1: Human evaluation results and corresponding automatic diversity metrics of the filtered passages with on-equal-footing fluency. Abbreviations of metrics include self-BLEU 4/5 (SB-4/5), Zipf coefficient (ZC), and 3-gram entropy (Ent-3)

| | |
|---|---|
| Human, PPL =28.25 | She walks in beauty, like the night \n Of cloudless climes and starry skies; \n And all that's best of dark and bright \n Meet in her aspect and her eyes; \n Thus mellowed to that tender light \n Which heaven to gaudy day denies. \n One shade the more, one ray the less, \n Had half impair'd the nameless grace \n Which waves in every raven tress, \n Or softly lightens o'er her face; \n Where thoughts serenely sweet express \n How pure, how dear their dwelling-place. (…) |
| Nucleus *p*=0.9, PPL =28.22 | She walks in beauty and strength and dresses all in white. Though her face may be so dry, it has not failed her either." \n At 10:30 p.m. Jack out in the deep blue area, pursued by shadow. \n B: You know this has been the same thing going on since the Lord was slain. His face always looks so brittle and breakable. He's so magnificent, like a hurricane of sorts; like glass and everything breaking into fragments around him. Still, I keep hoping to see his face, (…) |
| Top-*k* *k*=200, PPL =28.32 | She walks in beauty though she seems to bear a burden. \n\n Empire has no great enthusiasm for anything, but whenever our people wish to change their country's system, there is no rest for the wicked. So shall we then, being guilty of an error, even though it be wrong? We shall not then be blamed; there is no shame in what we try to do. These are all the laws that we invented by ourselves. From the beginning they proceeded independently. (…) |
| Temp. *t*=1.0, PPL =28.11 | She walks in beauty. "Love," some old man says, "Belongs to two constant as those two stars." Beautiful diagonal line. So beautiful, the trees try to straighten it. "Wait a second," Peter says. "Is this exactly the last one?" For an example, let's suppose it's the last blue smoke. "Our Remains," Peter says. "How in the Hell's name is that supposed to be a song, though . . . " Won't this just be boring, you ask. Sure, says Peter, (…) |
| IQR-IP *p*=0.8, *k*=640, PPL =28.27 (*ours*) | She walks in beauty through all things good, as though a prince in the bloom of youth were ever born in any city. For this I would never forget the time I had spent with her, when we went through this temple. The perfume, the beautiful woman, the silence, the strange shadows, the pleasant voice, the flower of every description, were like those which now from her new cell perfume the fair shrine of Venus." \n And his memory fades into sleep, for at this very moment Venus rises from her silent chamber. The Roman fable has the goddess emerging from her palace in an instant from the black night of death. When the men are searching for her she rises from her throne, where the eyes of Death watch her silently, to welcome them. And from her presence a tumult is born, a struggle in darkness, a terrible din, of discordant cries. For this reason it was always sung, that if any were in a black room they should hear the shrill sound |

Table 2: Examples with PPL near the reference passage (Lord Byron's poetry). These passages have similar PPL and can be viewed as similarly fluent. While traditional methods favor creating plain and narrative passages, our method generates novel and surprising passages on the same fluency level.

## 5 CONCLUSION

We propose the interquartile range inverse probability (IQR-IP) sampling algorithm. It rescales the high-likelihood part of the predicted distribution with inverse probability weighting to increase diversity and conducts multi-filtering truncation on the low-likelihood to preserve fluency. Results show that our algorithm can significantly increase the diversity and novelty of the generated text without corrupting the fluency. Our results suggest a method of manipulating the high-likelihood part of the predicted distribution to increase diversity and novelty. It might be beneficial for high-novelty cases such as poetry or music generation. Although superior to baselines, our method may be far from the optimal sampling algorithm regarding diversity and novelty issues. We believe there still exist undiscovered and better sampling algorithms for diverse open-ended neural text generation.

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

## A    DETAILS OF HUMAN EVALUATION

We use the following steps to construct Amazon Mechanical Turk (MTurk) human intelligence tasks (HITs). We first generate 5,000 passages following the same prompt "She walks in beauty" on all sampling parameter configurations for our algorithm and three baseline methods. Then we set five pre-defined PPL filters evenly distributed on the logarithmic PPL space since the quality judgment by the human is nearly linear to logarithmic PPL (Zhang et al., 2021; Basu et al., 2021). We ensure that the golden human-level PPL locates in the middle of the filters, and the PPL of Lord Byron's poetry locates in the rightmost. We also filter out passages that contain toxic words, non-English letters, or unreadable symbols. It results in 144 passages per PPL level per sampling algorithms, which makes $144 \times 5 \times 4 = 2,880$ passages in total for evaluation. We shuffle and split them into chunks containing ten passages as one HIT and set each HIT to be assigned to five different MTurk workers. The workers are asked to rate each passage on its fluency and novelty on a 1-5 scale (larger/better) within 10 minutes. The interface of the HIT is shown in Figure 14 and 15.

For quality control, we set the requirements for MTurk workers to be located in the US and have an acceptance rate no lower than 95%. We exclude responses from workers who spent less than 350 seconds on the HIT, i.e., less than 35 seconds on each passage. Following methods by Nadeem et al. (2020), we observe the convergence of the average human rate when pushing more HITs to MTurk. Under the proposed on-equal-footing fluency paradigm, we found that the average human rate converges quickly after 100 samples are evaluated, as is shown in Figure 16. We stop pushing HITs to MTurk shortly after the convergence of the human rate. In total, we collect annotations from 100 HITs by 500 workers, among which 166 workers' responses are included to report the human evaluation results in Table 1. The acceptance rate of MTurk workers' responses is 33%.

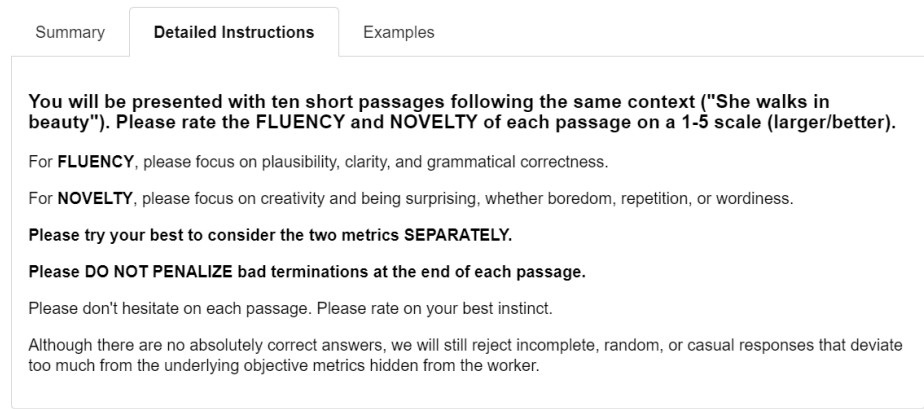

## Instructions

| Summary | **Detailed Instructions** | Examples |

**You will be presented with ten short passages following the same context ("She walks in beauty"). Please rate the FLUENCY and NOVELTY of each passage on a 1-5 scale (larger/better).**

For **FLUENCY**, please focus on plausibility, clarity, and grammatical correctness.

For **NOVELTY**, please focus on creativity and being surprising, whether boredom, repetition, or wordiness.

**Please try your best to consider the two metrics SEPARATELY.**

**Please DO NOT PENALIZE bad terminations at the end of each passage.**

Please don't hesitate on each passage. Please rate on your best instinct.

Although there are no absolutely correct answers, we will still reject incomplete, random, or casual responses that deviate too much from the underlying objective metrics hidden from the worker.

---

**Good examples**

**The timeless masterpiece by Lord Byron can be rated to have both HIGH FLUENCY and HIGH NOVELTY:**

She walks in beauty, like the night
Of cloudless climes and starry skies;
And all that's best of dark and bright
Meet in her aspect and her eyes
Thus mellow'd to that tender light
Which heaven to gaudy day denies.

One shade the more, one ray the less,
Had half impair'd the nameless grace
Which waves in every raven tress,
Or softly lightens o'er her face;
"Where thoughts serenely sweet express
How pure, how dear their dwelling-place

And on that cheek, and o'er that brow,
So soft, so calm, yet eloquent,
The smiles that win, the tints that glow,
But tell of days in goodness spent,
A mind at peace with all below,
A heart whose love is innocent!

**Bad examples**

**A passage with LOW FLUENCY might be unreasonable and hard to read:**

She walks in beauty and enjoyment for a height of 5'11 along the beach boardwalk separating the more expensive beach front and the pleasant walking beach based around the substantial right bank coffeesite.
Established in what used to be storied Gucci Town, Ms. Moore went to dinner restaurant route after work. What began as a tiring day would become a day of bliss. Mr. Zlola James's shares in the two door after work restaurant hopped from the dollar to the dollar, incorporating the idea of constant business.

**A passage with LOW NOVELTY might be repetitive, wordy, and boring:**

She walks in beauty with her long golden locks, her green eyes, her delicate figure, her sensuous and beautiful voice, her beautiful face, her beautiful body, her beautiful smile, her beautiful eyes. She is the most beautiful woman in the world, and she is only a child.
"You are the most beautiful woman in the world, and you are only a child."
We are not children, we are the most beautiful women in the world.
"You are the most beautiful woman in the world, and you are only a child."
I know that I am the most beautiful woman in the world, and I am only a child.
"You are the most beautiful woman in the world, and you are only a child."

**NOTE**

**Although boring and wordy, the second passage is grammatically fluent. It can be rated to have HIGH FLUENCY but LOW NOVELTY.**

**On the contrary, the first passage is surprising and informative. It can be rated to have LOW FLUENCY but HIGH NOVELTY.**

**Please try your best to consider the two metrics SEPARATELY.**

Figure 14: Instruction of the Amazon Mechanical Turk (MTurk) human intelligence tasks (HITs) for our human evaluation experiment.

Interestingly, we find that the threshold of workers' time consumption on the HIT affects the prominence of advantage for our method. As is shown in Figure 17, a lower and less strict threshold of 250 seconds diminishes the gap between our method and baseline methods, while a higher and more strict threshold of 350 seconds (which is the case in Table 1) brings more prominence to the advantage of our method. It reveals that the high-quality annotations by workers who spend more time working on the task favor our method over baseline methods.

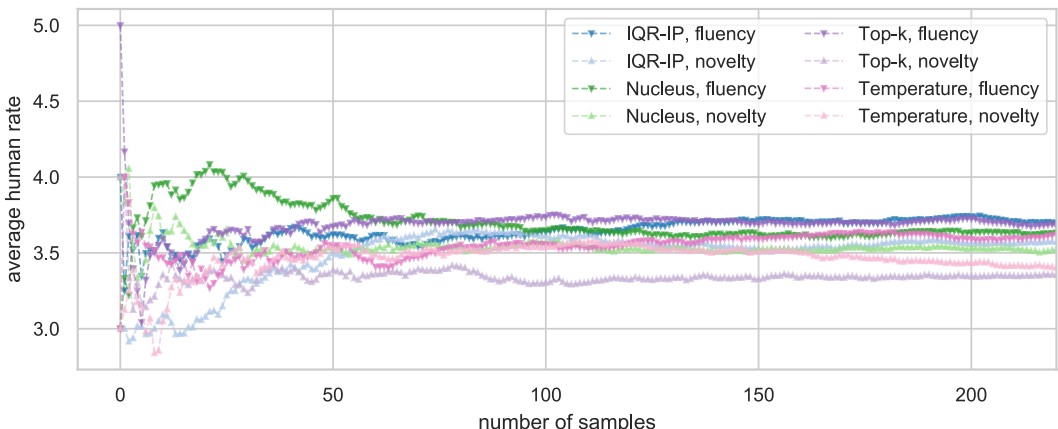

Figure 15: An example questions from the HIT.

Figure 16: Convergence of average human rate when increasing the number of samples evaluated by MTurk workers.

We further illustrate the details by aligning human rates against PPL in Figure 18. The results show that our method has a great advantage for the fluency metric on the highest PPL level and has stable advantages for the novelty metric on most PPL levels, which results in the highest overall rate. It could be explained that the high PPL of our algorithm could possibly be contributed by the rescaled high-likelihood words on the "head" rather than filtered-in low-likelihood words on the "tail", which results in a higher fluency rate on the high PPL level. On the other hand, since the rendered word distribution of our method is always flatter than baseline methods, the passages on each PPL level can always achieve a high novelty rate.

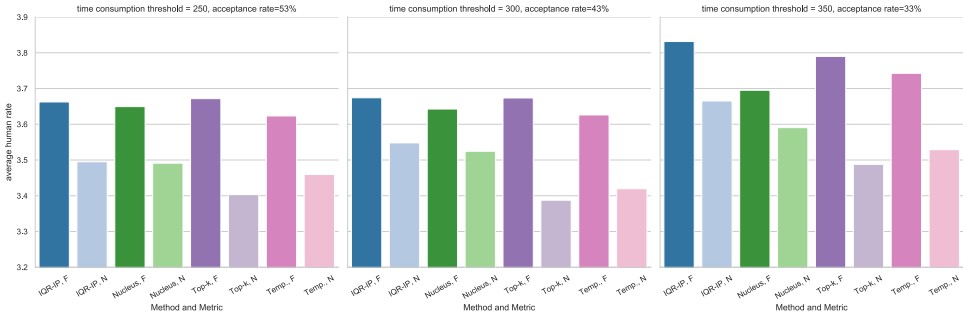

Figure 17: Human evaluation results are slightly different by tuning the filtering thresholds of time consumption on the HIT responses (from left to right are 250, 300, and 350 in seconds). Increasing the threshold (lowering the acceptance rate of MTurk workers' response) brings more prominence to the advantage of our method and vice versa. Abbreviations of metrics include fluency (F) and novelty (N).

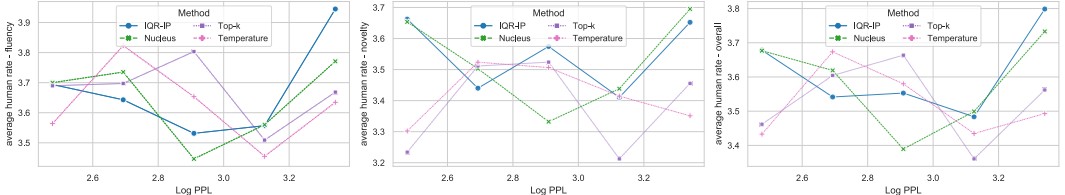

Figure 18: Results of human rate for fluency/novelty/overall against PPL.

## B  THE EXTENDED DIVERSITY BOUNDARY

We provide total variance analysis to explain the expansion of the diversity boundary of our algorithm. Following proposition by Kang & Hashimoto (2020), we can evaluate the upper bound of total variance between $p_{inv}(x)$ and reference distribution $p_{ref}(x)$ with the following corollary.

**Corollary 1.**  Upper bound of total variance between $p_{inv}$ and $p_{ref}$ satisfies

$$|p_{inv} - p_{ref}|^2 \leq \frac{1}{2} KL(p_{ref}||p'_{fil}) + 2m + m^2, \tag{6}$$

where

$$m = \max_{x \in V^{VeryHigh}} |p'_{fil} - \frac{Z_p}{p'_{fil}}|, \tag{7}$$

$$Z_p = \frac{\sum_{x \in V^{VeryHigh}} p'_{fil}}{\sum_{x \in V^{VeryHigh}} p'_{fil}^{-1}}. \tag{8}$$

*Proof.* First, with Pinsker's inequality (Csiszár & Körner, 2011), the total variance between the original filtered distribution $p_{fil}$ and the reference distribution $p_{ref}$ satisfies

$$|p'_{fil} - p_{ref}|^2 \leq \frac{1}{2} KL(p_{ref}||p'_{fil}). \tag{9}$$

Then we may use similar methods by Kang & Hashimoto (2020) to derive the new bound as follows.

$$|p_{inv} - p_{ref}|^2 \leq (|p_{inv} - p'_{fil}| + |p'_{fil} - p_{ref}|)^2 \tag{10}$$

By definition of $p_{inv}$ in Equation 5, we have

$$|p_{inv} - p'_{fil}|^2 \leq \max_{x \in V^{VeryHigh}} |p'_{fil} - \frac{Z_p}{p'_{fil}}|. \tag{11}$$

Then expand Equation 10, and use $m$ defined in Equation 7 and 11 to bound $|p_{inv} - p'_{fil}|$, and use Equation 9 to bound $|p'_{fil} - p_{ref}|$, the inequality is proved.  □

Equation 6 reveals an additional term controlled by $m$ besides the original bound $\frac{1}{2}KL(p_{ref}||p_{fil})$ (achieved by $p'_{fil}$ without inverse probability weighting). Since $m$ contains a value of inverse probability, the new upper bound will change dramatically, which provides a diversity enhancement measure.

The corollary has the same form as Kang & Hashimoto (2020), although with a different constant $m$, which corresponds to the truncation ratio $c$ of their proposition. In our work, $m$ is controlled by inverse probability weighting and can be reasonably large, while the truncation ratio $c$ satisfies $0 \leq c \leq 1$. In this way, it can be regarded as an extension of the proposition by Kang & Hashimoto (2020) with unbounded $c$.

Note that since $0 < Z_p \leq 1$, $\max |p'_{fil} - \frac{Z_p}{p'_{fil}}|$ can only be achieved on the largest or smallest value of $p'_{fil}$ in $V^{VeryHigh}$, i.e., on the first or last word of $V^{VeryHigh}$. As a result, $m$ is controlled by $\rho$ in Equation 2 and filtering parameters in Equation 4. For example, with a loosely filtered $V'_{fil}$, $V^{VeryHigh}$ might contain the last word with a too-small probability and render a too-large value of $m$. Hence the total variance will become too large and corrupt the fluency. However, with carefully chosen parameters, $m$ may provide reasonable diversity enhancement without hurting the fluency, as is shown in the evaluation results.

## C  METRIC VARIATION FOR IQR COEFFICIENT AND TOP1CTRL FILTERING

| Method | PPL | SB-4 $\downarrow$ | SB-5 $\downarrow$ | ZC $\downarrow$ | Ent-3 $\uparrow$ |
|---|---|---|---|---|---|
| IQR-IP, $n = 100$, $\rho = 1.5$ | 16.77 | 0.47 | 0.29 | 1.17 | 5.24 |
| $\rho = 3.0$ | 14.90 | 0.50 | 0.32 | 1.22 | 5.22 |
| $\rho = 5.0$ | 12.76 | 0.52 | 0.34 | 1.26 | 5.19 |
| $\rho = 10.0$ | 11.57 | 0.53 | 0.36 | 1.39 | 5.18 |
| $\rho = 50.0$ | 9.62 | 0.55 | 0.39 | 1.54 | 5.09 |
| $n = 10$ | 13.39 | 0.53 | 0.35 | 1.22 | 5.21 |
| $n = 50$ | 16.50 | 0.48 | 0.30 | 1.17 | 5.23 |
| $n = 200$ | 19.48 | 0.45 | 0.28 | 1.15 | 5.24 |
| $n = 1000$ | 20.52 | 0.44 | 0.27 | 1.15 | 5.24 |

Table 3: Metric variation for IQR coefficient $\rho$ and Top1CTRL filtering parameter $n$.

We present the metric variation by tuning the IQR coefficient and Top1CTRL filtering. We choose to start from the sampling parameter configuration with $k = 640, p = 0.8$ (fixed) which locates nearest to the reference point on the Q-D plane, then tune $\rho$ and $n$ away from their originally fixed values to observe the variation of automatic metrics, as is shown in Table 3. When $\rho$ in Equation 2 increases, it shortens the identification range of $V^{VeryHigh}$. Hence it decreases the intensity of inverse probability weighting, lowering the diversity metric. If $\rho$ is increased to infinity, there will be no $V^{VeryHigh}$ and our algorithm will degrade to plain stochastic sampling filtered by Equation 1 to 4.

For Top1CTRL filtering, Table 3 shows that loosening $n$ exhibits the same behavior to loosening $p$ and $k$. Note that the metric does not significantly vary when $n \geq 100$, since the scale constraint becomes weak. On the other hand, our fine-grained mechanism in Equation 4 ensures that a small value of $n$ does not over-prune the vocabulary, which is reflected by the smooth variation of metrics when $n \leq 100$.

We argue that the sampling parameters $(p, k, n, \rho)$ can be fixed around the reference point since the fixed parameters already yield a distribution of PPL that is very close to human-level metrics, as is shown in Figure 13. We also argue that controlling the PPL of generated passages can be handled using either the post-decoding filtering method in our experiment or the dynamic parameter methods

such as MIROSTAT (Basu et al., 2021), rather than cumbersome hyper-parameter tuning. This is an orthogonal topic, so we leave it for future work.

One issue to clarify is that by the definition of IQR, there should be a $V^{VeryLow}$ that locates symmetrically to $V^{VeryHigh}$ on the identification range in Figure 5. Our experiment suggests that this boundary is always below 0, i.e., $V^{VeryLow}$ is always an empty set during IQR calculation. As a result, we omit the description for $V^{VeryLow}$ in Equation 2.

Note that one may even design different rescaling strategies besides Equation 5, e.g., evenly redistributing $V^{VeryHigh}$, or add some noise on $V^{VeryHigh}$, to achieve a less severe variation bounded by Equation 11. In that case, our algorithm is an extreme case that we try to leverage the human judgment quality-likelihood curve by Zhang et al. (2021) and re-order $V^{VeryHigh}$ entirely with inverse probability weighting.

