# OpenReview forum: "Penalizing the High-likelihood: A Novel Sampling Method for Open-ended Neural Text Generation via Inverse Probability Weighting"
_ICLR.cc/2023/Conference — Submitted to ICLR 2023_

### Official Review · Reviewer_yKG8 · 2022-10-23

**Confidence:** 4
**Correctness:** 2
**Technical Novelty And Significance:** 2
**Empirical Novelty And Significance:** 3
**Recommendation:** 3

**Clarity, Quality, Novelty And Reproducibility:**

A few other comments
- The design decision of Equation (1) is not totally clear to me.
- The design decision of Equation (2) seems quite arbitrary to me. How is rho chosen?
- What does “regularized” mean in “regularized distribution on V” under Equation (1)?


**Strength And Weaknesses:**



Decoding is an important problem, and I’m glad that there’s attention on this issue.

It’s good to see that the authors have pointed out that lower perplexity does not equal better quality. It’s good to see the human evaluation details.


The likelihood trap isn’t a strong enough motivation for the authors’ algorithm.
- First, the authors point out the likelihood trap where the “repetition tendency grows stronger when more loops occur” (which the degeneration paper by Holtzmann et al. already discussed) – if cognitive scientists do the same experiments for humans, it’s likely that humans will also assign higher probabilities to repetition loops if there are already repetitions in the prefix.
- Second, to solve this problem, one simple strategy is n-gram blocking, so we just don’t generate a sentence with two trigrams that are exactly the same, for example. See Paulus et al. (2018) https://openreview.net/pdf?id=HkAClQgA- (Section 2.5) and Chen et al. (2021) https://arxiv.org/pdf/2012.14919.pdf (search n-gram blocking).

On empirical results…
- The authors only experimented on language modeling (unconditional text generation). I’m not convinced that this approach would work on the vast majority of text generation applications (translation, summarization, question answering, or even dialogue).
- It’s not to me whether the proposed approach will work for longer story generation.
- It’s not clear to me whether human evaluation considers the consistency of the generated article.
- For baselines, it’s not clear how the k, p, temperature are tuned. It’s also not clear whether the same human evaluation trend will work for other perplexity levels.
- What is the annotator agreement? Please let me know if I missed this information.
- Relatedly, the authors claim (at the end of page 4) that V^{very high} will be singleton on a peaked distribution that contains unquestionably right words. But it’s not clear to me what the boundary between “unquestionably right” and “questionably right” is.

It’s a bit unconvincing to me that the authors’ claim that “lower likelihood words on a flat distribution are reasonable choices” in Section 2.2 is only verified on one example only (Figure 4).

Relatedly, the authors may have misrepresented previous work in Section 2.2. The authors claim that “it is proven beneficial to increase generation diversity by emphasizing less probable words during training” and have cited Welleck et al. (2020). But the unlikelihood objective in the Welleck et al. paper, in fact, is trying to make incorrect repeating tokens less likely and make frequent tokens less likely. This is not equivalent to simply making less probable words more likely.


**Summary Of The Paper:**

The paper claims that humans do not always favor high-likelihood texts; decoding using high likelihood leads to repetition and boredom. So, the paper proposes a sampling method. It rescales the high-likelihood probabilities in a distribution (determined by quartiles and interquartile range aka IQR, as shown in Equation 2). The paper claims that the rescaled distribution has a “closer resemblance to the quality-likelihood curve” shown in Figure 1. Experimenting on unconditional text generation (using a specific prompt) – generation is a-few-sentence long, novelty and diversity are better for this approach compared to the baselines.


**Summary Of The Review:**

Interesting observation that sometimes we don't want highest-scoring tokens, but algorithm design decisions are sometimes arbitrary. Experiments are limited on a specific task. Baselines are not entirely convincing.

---

> ### Author Response · Authors · 2022-11-16
> **Response to Reviewer yKG8**
>
> We thank the reviewer’s very valuable and insightful comments.  We make the following explanations.
>
> -- Weakness 1. We have to argue that the purpose of this work is not to handle or eliminate the likelihood trap phenomenon, which we totally agree can be easily handled by simple rules such as n-gram blocking or even a penalty on the repetitive tokens from the context. The true purpose is to testify a new counter-intuitive rescaling mechanism on the seemingly always trustworthy high-likelihood tokens. The intuition of such a method comes from the observation of the likelihood trap. Actually, our results in Section 4 implied that the likelihood trap was not “eliminated”, for example, the 3-gram entropy of our algorithm may still approach a trapped state with a small value (see Figure 10). We think the study of the likelihood trap was not at all the core contribution of this work but rather an inspiration to derive the manipulations. And as we stated in Section 4.2, our method violates the common properties (entropy reduction property or slope preservation property [1]) of decoding algorithms but results in better behavior (closer to the human point on the trade-off plane, and higher novelty score by human evaluation with similar fluency level). Such a counter-intuitive method can be important to understand the novelty gap for the generation task.
>
> -- Weakness 2
> 1. We would like to argue that open-ended generation is fairly unique compared to other tasks. It requires automatic metric evaluations as well as carefully designed human evaluations. Due to the limitation of space, we hope to fully cover and testify the validity of our approach like existing works [1-4] which all choose to evaluate on open-ended generation cases. And since our results show that the proposed method has similar performance to traditional methods (top-p/k) regarding fluency, we think it is implied that it can be used on other generation tasks that mainly focus on the fluency metric.
> 2. For the long-story generation issue, we think it is rather an orthogonal topic for decoding algorithms since none of the existing algorithms can guarantee ubiquitously good performance on such a challenging task.
> 3. For the consistency issue in human evaluation, we believe it is yet another paralleled topic out of the scope of decoding algorithm research. The well-known and most concerning issue is to trade off fluency and novelty, so we only set those two metrics for human evaluation.
> 4. The process of tuning p/k/t to filter the passages for evaluation is a core part of our experiment. We state clearly that due to the quality variation describe in Figure 13, it is important to create a fair comparison for all methods that address the variation issue. We also explained the choice of perplexity levels in Appendix A, which can also be inferred from Figure 13 that these levels cover the major part of the distribution for quintessential methods/parameters such as top-p=0.90 and 0.95.
> 5. The annotator agreements as well as quality control mechanisms are described in detail in Appendix A.
> 6. The “unquestionably correct” case refers to the peaked distribution such as Figure 5b as well as many previous works such as [4].
> 7. Figure 4 only serves as an intuition to derive the manipulation. Its validity is supported by all results in Section 4 and Appendix.
> 8. On the reference issue to Welleck et al, you kindly state that “trying to make incorrect repeating tokens less likely and make frequent tokens less likely” is not “equivalent to simply making less probable words more likely”. We have to disagree with this statement and argue that in the spirit of probability sampling, a suppression of specific tokens will in return result in an emphasis on other tokens, since the manipulated distribution is practically and must be NORMALIZED, after which the decreased amount of probability mass is assigned to other parts of the space, as is described in Eq. 5.
>
> -- On other comments.
> 1. As we stated in Section 3.1, Eq. 1 serves to perform fine-grained filtering on the low-likelihood tokens.
> 2. The choice of setting $\rho$ to 1.5 is a common method for calculating interquartile range (IQR) value. Regardless, in Appendix C we also have attached results by tuning $\rho$ which serves as the ablation study of weakening the rescaling on the high-likelihood tokens.
> 3. We apologize that the word “regularized” is a faulty choice in which we meant a normalization approach on a probability vector. We would like to fix it into “normalized” in the updated draft.
>
> Again, we thank the reviewer’s valuable comments. We hope our replies could clarify your concerns on the paper.
>
> [1] Nadeem et. al., A systematic characterization ...
> [2] Zhang et al., Trading off diversity and quality ...
> [3] Basu et al., MIROSTAT. ...
> [4] Holtzman et. al., The curious case ...

---

> > ### Comment · Reviewer_yKG8 · 2022-11-29
> > **Partial response to authors' rebuttal**
> >
> > Appendix A and Figure 17 are really helpful. I also wonder about the annotator agreement (like Fleiss' kappa or other metrics).
> >
> > On response 2-8: Welleck et al. only showed that "making less probable words more likely" would work, when the "less probable words" are non-repeating words. So I would qualify/restrict the original sentence in the paper.
> >
> > I'm still going through other comments. Thanks authors for the response!

---

### Official Review · Reviewer_bnch · 2022-10-25

**Confidence:** 4
**Correctness:** 3
**Technical Novelty And Significance:** 2
**Empirical Novelty And Significance:** 2
**Recommendation:** 3

**Clarity, Quality, Novelty And Reproducibility:**

Clarity

I believe the paper gets the message across, but could have a much clearer motivation.

Quality

The work is overall of medium quality. The motivation for the method is muddled. The experiments are quite systematic in terms of mechanical turk investment, etc, but seem to have a glaring flaw with only one single prompt.

Novelty

The method is mostly just a mix of previous methods, with the exception of the IP part. As far as I am aware, that is novel in this context.

Reproducibility

Full code is provided, so theoretically it is 100% reproducible.



**Strength And Weaknesses:**

Strengths

+ The paper is reasonably clear, although certain parts are hard to understand exactly.

Weaknesses

+ The method is not very well justified. The authors do not clearly describe why their method is appropriate. Their central idea is that of inverse-probability weighting the 'head' of the candidate distribution. This is justified with a vague reference to [Zhang 2021] and 'causal inference', but I think this central point needs to be justified better. For instance, if I have a 'head' with probability [1/2, 1/3], after the IQR-IP algorithm I have candidates with probability [1/3, 1/2]. After reading the paper closely, it still seems somewhat mysterious why this would make sense.
+ The experiments are relatively weak. The most glaring flaw is that as far as I can tell, all the experiments are carried out with the same prompt ('She walks in beauty'). This doesn't give me much confidence that the results are going to generalize to other prompts. Similarly, the comparisons are lacking. Given the structure of IQR-IP, it would make sense to have a comparison with the intersection of top-p and top-k. Furthermore, the experiments only use a single, relatively small model (GPT2-XL). It is common practice to evaluate sampling methods over several models, particularly paying attention to the change in performance as the number of parameters increases. The authors could use the recently-released OPT series of models to study this.
+ Nitpick: Why is there no V^VeryLow?

**Summary Of The Paper:**

The paper proposes a new probabilistic decoding method for generating a sequence of text from a trained autoregressive model $p(x_{t+1}|x_{1:t})$. The method, IQR-IP, is carried out by first doing top-p and top-k filtering on the possible generations, then computing the quantiles of the remaining tokens. For quartiles $Q_1, Q_2, Q_3, Q_4$, we take the set of generations with probability greater than $Q_3 + \rho \cdot\text{IQR}$, where $\text{IQR}$ is the interquartile range. Then we reweight this set of candidates. This reweighted set is combined with the post top-k, top-p set to get the set of candidates, which are sampled from.

**Summary Of The Review:**

The paper proposes a method which appears to have some slight advantages over competing decoding methods. However, since it is only evaluated on one prompt and contains several ad-hoc choices of filters, parameters etc, it's difficult to say if the method would generalize to other prompts. Similarly, we only know the performance on a single model, which is relatively small in 2022.

---

> ### Author Response · Authors · 2022-11-16
> **Response to Reviewer bnch**
>
> We thank the reviewer’s very valuable and insightful comments. We make the following explanations.
>
> -- Weakness 1 (on the inverse probability weighting). Due to the limitation of space, we only made a simple reference to the conclusion from causal inference. The conclusion demonstrates an inverse correlation between the importance of an event and its probability. In your example, the vector of [1/2, 1/3] is rescaled to [2/5, 3/5], not just reversing their order like [1/3, 1/2]. And although this is an intuitive method, its advantage is thoroughly justified by our experiment.
>
> -- Weakness 2 (experiment settings). On the single prompt issue, this is rather an approach to guarantee the reproducibility of the human evaluation results. As is widely acknowledged, the research of sampling methods for open-ended neural generation is notoriously difficult to reproduce. As can be seen in numerous works such as in [1-4], different prompt for human evaluation easily results in huge disagreement among annotations. For this purpose, we testify to the possibility of using a single prompt to create an easier and clearer task for human evaluation. As is shown in the instructions of MTurk HIT in Figure 14, the sharing prompt creates a highly comparable scenario that makes MTurk workers easy to annotate, which creates faster convergence of human rate in Figure 16, compared to results in [3]. With this huge convenience, we directly set automatic evaluation to use a single prompt as well to guarantee consistency of evaluation. The success of this approach can also be inferred from Table 1, where the golden 3 traditional methods actually achieve an identical overall score, which follows the results from [3] that these baseline methods actually have similar performance. The comparison between our method and the intersection of top-p/top-k was presented in Appendix C, where when \rho approaches a large value, the rescaling becomes weak and practically degrades into the intersection of the top-p/top-k case. On the big-model issue, many existing works dealing with the sampling method use the GPT2 family, and we hope to provide comparable results with their works by using a similar type of model. In the future, we are interested to testify the results on a larger model.
>
> -- Nitpick. We greatly appreciate the review’s rigorous examination of our approach. Actually, we specifically mention the $V^{VeryLow}$ issue in Appendix C on Page 16 that it “is always an empty set during IQR calculation”. We apologize that due to the limitation of space, these parts are not presented in the main sections.
>
> Again, we thank the reviewer’s highly rigorous and professional comments. It is our delight to learn that the reviewer has conducted a careful examination of our algorithms. Many of the reviewer’s concerns can be found in the Appendices. We hope our replies could clarify your concerns.
>
>
> [1] Zhang et al.,  Trading off diversity and quality in natural language generation.
> [2] Basu et al., MIROSTAT: A neural text decoding algorithm that directly controls perplexity.
> [3] Nadeem et. al., A systematic characterization of sampling algorithms for open-ended language generation.
> [4] Holtzman et. al., The curious case of neural text degeneration.

---

### Official Review · Reviewer_yD5T · 2022-10-25

**Confidence:** 4
**Correctness:** 2
**Technical Novelty And Significance:** 2
**Empirical Novelty And Significance:** 2
**Recommendation:** 3

**Clarity, Quality, Novelty And Reproducibility:**

While not yet published in a venue (and therefor doesn’t have an impact on my assessment) I believe in the future authors should compare their approach to prior work by (Meister et al 2022: ****Locally Typical Sampling****) since it introduces the similar idea of pruning the head of the distribution to increase diversity, but with much less hyperparameters and more theoretically motivated.

**Strength And Weaknesses:**

- The analysis earlier on in the paper is interesting. Although the repetitive-ness problem has been identified before, the connections made with high-likelihood tokens and their increasing probability is relevant and should be highlighted.
- I believe that modifying the sampling algorithm to penalize tokens in the "head" is a promising direction to improve the quality of samples decoded from large models.

I think the paper has a couple of important flaws:

- The proposed method lacks throughout analysis and isn’t particularly novel or theoretically motivated. Half of the procedure is just applying two well known pruning techniques, nucleus and top-k. The choice of only re-scaling the top quartile seems kind of arbitrary and not well studies (why not pick, lets say, the top 10% percentile?) and there is very little discussion on why rescaling according to their inverse probability values is a good choice. Given that this algorithm is very heuristic, these choices need to be motivated by having a good ablation study.  Also, their proposed sampling algorithm has 3 hyperparameters, leading me to believe that it will be hard to tune in practice
- The experimental setup is lackluster, and the results are disappointing. The authors only evaluate their algorithm according to generated completions for a **single** prompt and by a **single** model, compare to other simple baseline sampling algorithms for the same prompt and model. The results then show that their model achieves comparable perplexity to human references and a bit more novelty according human-references. However the differences in values are very small, and since no confidence bounds are statistical tests are done, it’s hard to grasp if their method is clearly better



**Summary Of The Paper:**

This paper proposes a novel sampling algorithm for distributions learned by neural language models. The authors start by highlighting a pitfall of “naively” sampling from these models, where high likelihood tokens being picked leads to their future probability increasing and to repetitive loops in the samples.

To increase diversity, they then propose modifying the distribution we sample from by rescaling this high-likelihood tokens. After the removal of tail, very low-probabity tokens with previously proposed methods (such as top-k and nucleus), the authors propose rescaling the tokens belonging to the top quartile (25%) of probabilities to make their probabilities proportional to their inverse.

To evaluate their propose sampling algorithm, a pretrained GPT-2 XL model is used to generate samples from a pre-defined prompt:

“She walks in beauty”

generating 5000 sample passages for their method and baseline sampling approaches. They then evaluate this passage according to (1) fluency based on the perplexity of the model (2) diversity based on Self-BLEU, N-gram entropy and Zipf coeficient and (3) 5-point Likert Scale for Fluency & Novelty. Finding that their method proposes comparably to other methods in human evaluation, arguably generating more diverse samples.

**Summary Of The Review:**

This paper proposes a novel sampling algorithm that down-weights the probabilities of the most-likely tokens in the distribution. While the intuition behind it is interesting and has potential, some of the design choices are not well-explored and the experimental evaluation is very lack-luster.

---

> ### Author Response · Authors · 2022-11-16
> **Response to Reviewer yD5T**
>
> We thank the reviewer’s very valuable and insightful comments. We make the following explanations.
>
> -- Weakness 1 (the heuristic part). We admit that many parts of algorithm designs are heuristic and by intuition, e.g., the inverse probability (IP) approach, and the 25% percentile approach. However, as we have mentioned in Section 2.2, IP comes from the results of causal reference which re-weights the probability of an event with its inverse value. For the rescaling scheme, there can be too many manipulations as long as it has an inverse correlation with the likelihood, which could be intractable. In our work, we only wish to raise the point that an inverse manipulation on the high likelihood is counter-intuitively helpful. But which manipulation might be optimal remains a difficult question to answer. Consequently, we only follow one of the widely acknowledged conclusions from the field of causal inference that uses IP manipulation. For the percentile issue, the value of 25% comes from the standard setting of interquartile range (IQR) calculation. For the ablation study. We have presented such results in Appendix C, in which a greater value of \rho suggests a weaker rescaling effect which gradually degrades into the traditional filtering method. As for the hyperparameter issue, we also mentioned in Appendix C that these parameters should be fixed around the reference point. Neither do we think that hyperparameter tuning is a good way to control the generation, as we present the quality variation issue in Figure 13.
>
> -- Weakness 2
>
> --- On the single prompt issue, this is rather an approach to guarantee the reproducibility of the human evaluation results. As is widely acknowledged, the research of sampling methods for open-ended neural generation is notoriously difficult to reproduce. As can be seen in numerous works such as in [1-4], different prompt for human evaluation easily results in huge disagreement among annotations. For this purpose, we testify to the possibility of using a single prompt to create an easier and clearer task for human evaluation. As is shown in the instructions of MTurk HIT in Figure 14, the sharing prompt creates a highly comparable scenario that makes MTurk workers easy to annotate, which creates faster convergence of human rate in Figure 16, compared to results in [3]. With this huge convenience, we directly set automatic evaluation to use a single prompt as well to guarantee consistency of evaluation. The success of this approach can also be inferred from Table 1, where the golden 3 traditional methods actually achieve an identical overall score, which follows the results from [3] that these baseline methods actually have similar performance. For the single model issue, many existing works [1-4] have used a single model to testify their decoding algorithms. We seek to follow their paradigm for comparable results.
>
> --- On the significance of the advantage of our method, please see Appendix A in which a stricter quality control threshold (thresholds of time consumption on the HIT responses) brings more significance. We have presented the results in Figure 17 that when we raise the threshold up to 350 seconds the significance becomes much more prominent. It indicates that our method is favored by rigorous MTurk workers who spent more time on the HIT. Due to the limitation of space, we only put results with 300s as the threshold in Table 1, which might have created an impression of insignificant results. This could be easily fixed by either typing all the numbers from Figure 17 into Table 1 given more spaces, or simply replacing them with the 350s case in Figure 17. We will upload a new version of the draft in which Table 1 presents the results using 350s as the threshold, which shows more prominence of the significance:
>
> | Method | Fluency | Novelty | Overall |
> | --- | ----------- | ----------- | ----------- |
> | Nucleus | 3.70 | 3.59 | 3.65 |
> | Top-*k* | 3.79 | 3.49 | 3.64 |
> | Temperature | 3.74 | 3.53 | 3.64 |
> | IQR-IP (***ours***) | **3.83** | **3.67** | **3.75** |
>
> -- On the choice of baseline methods. We have made a few statements in the response to Reviewer XtRt (--Weakness 2) on this issue. Although we do think it is unquestionably correct to compare with more algorithms, we think the comparison with the three commonly used golden methods like existing works [1-4] could already make the points of our work.
>
> Again, we thank the reviewer’s highly rigorous and professional comments. We hope our replies could clarify your concerns.
>
>
> [1] Zhang et al.,  Trading off diversity and quality in natural language generation.
> [2] Basu et al., MIROSTAT: A neural text decoding algorithm that directly controls perplexity.
> [3] Nadeem et. al., A systematic characterization of sampling algorithms for open-ended language generation.
> [4] Holtzman et. al., The curious case of neural text degeneration.

---

### Official Review · Reviewer_XtRt · 2022-10-28

**Confidence:** 5
**Correctness:** 3
**Technical Novelty And Significance:** 2
**Empirical Novelty And Significance:** 3
**Recommendation:** 5

**Clarity, Quality, Novelty And Reproducibility:**

Please see my review above. The paper can be made clearer, especially with more context about [1].

Regarding novelty, the issues with autoregressive models and open-ended generation are well documented in the prior work and some attempts like [2] have also attempted to rescale the distribution such that the high likelihood words are suppressed during generation. The proposed approach is novel in a precise sense because the exact algorithm has not been proposed before to the best of my knowledge. However, this work needs a more thorough comparison with other existing work in this area.

The approach is straightforward to implement and hence it should not be difficult to reproduce the results.

**Strength And Weaknesses:**

Strengths:

-- This paper identifies valid concerns with the quality of sampling from autoregressive models for open-ended generation and the scheme is motivated by empirical findings in recent works studying generation from large autoregressive models.

-- The empirical comparison includes various sensible metrics and comparisons which give a holistic picture of the effect of the proposed scheme.

-- The proposed scheme is straightforward to implement and is not any more computationally expensive than other common methods used for ancestral sampling-based schemes.

-- The results suggest that this approach performs comparably to other common sampling approaches, but has higher diversity than approaches like Nucleus sampling which is encouraging but not surprising given the scheme's effect of flattening the distribution.

Weaknesses:

-- The scheme does not seem grounded in any statistical principles related to sampling from probabilistic generators and neither does it ground itself in a clear objective that needs to be optimized. It involves arbitrary heuristics and introduces many more hyperparameters which seem crucial to get right from the results.

-- This paper fails to discuss and compare with other related work that also rescales the distribution such that the "head" is suppressed compared to other items in the vocabulary. An example of such work is [2], that uses entropy of the token-level distribution to rescale the distribution. Empirical comparison should be made against such works as well. Also, although cited, more discussion about comparison with other related work like [3] should be done in the main body.

-- This paper refers to [1] at many places but doesn't sufficiently describe the findings and setup of that paper which makes it difficult to read as a standalone paper.

Moreover Figure 1 of [1] refers to human judgements over full sentences whereas this work seemingly incorrectly applies that finding to *token-level distribution* in their argument (Figure 1 of this paper).

-- The human judgement section in the main body is difficult to understand.

[1] Trading off diversity and quality in natural language generation. Zhang et al., 2021
[2] Locally Typical Sampling. Meister et al., Feb 2022
[3] MIROSTAT: A neural text decoding algorithm that directly controls perplexity. Basu et al., 2021

**Summary Of The Paper:**

This paper proposes a scheme to improve open-ended generation from autoregressive models that is based on manipulating the token-level distribution of the model during ancestral sampling. Specifically, this scheme proposes pruning off the tail at each step like other popular decoding methods like Nucleus sampling and top-k sampling. But in addition to pruning, this scheme also involves rescaling the "head" the distribution so that items with very high likelihoods are deliberately suppressed compared to other items in the vocabulary in order to improve diversity and reduce repetition in the generated text. For this heuristic rescaling, the distribution over the vocabulary is divided into bins and the bin with the highest likelihood is reweighted according to the inverse probability of the items. This scheme is empirically compared to other popular approaches like Nucleus sampling, top-k sampling on various metrics like fluency, diversity, human judgement etc.

**Summary Of The Review:**

The paper is generally well- written and the approach is reasonable and is supported by clear experiments. However, comparison to other existing work is inadequate and the approach seems seems dependent on arbitrary heuristics.

---

> ### Author Response · Authors · 2022-11-16
> **Response to Reviewer XtRt**
>
> We sincerely thank the reviewer’s valuable and insightful comments. We make the following explanations.
>
> -- Weakness 1 (lack of principles and optimization). Due to the limitation of space, we only conduct a few theoretical derivations in Appendix B which helps to explain the extended diversity boundary. For the optimization issue, if you refer to the formats like Eq. 10 to 12 in Locally Typical Sampling [2] by Meister et al., we think we can easily change the filtering process of our algorithm into such formats. However, we don’t think it is helpful since our algorithm contains complicated manipulations on the distribution which is already hard to grasp. We sequentially present these steps from Eq.1 to 5 to help the readers to comprehend the process. Converting them into an “optimized” form where the filtered vocabulary satisfies constraints from Eq.1 to 4 will make it much more difficult to read. We think such a difference in presentation does not affect the validity of our method. On the hyperparameter issue, we mentioned in Appendix C that these parameters should be fixed around the reference point, following conclusions from many existing works [4,5]. We don’t think that hyperparameter tuning for our algorithm is necessary since results in Figure 13 already show that ALL concerned sampling algorithms exhibit distributional and variational quality, and controlling such variation demands paralleled techniques rather the hyperparameter tuning. We hope this could clarity your concerns.
>
> -- Weakness 2 (other baselines). You have kindly mentioned Typical Sampling by Meister et al. and MIROSTAT [3] by Basu et al. To our very limited comprehension, those two methods do not RESCALE the distribution of the high-likelihood words, but rather conduct adaptive TRUNCATION/PRUNING (NOT RESCALING) on the vocabulary, i.e., the (high-/low-) likelihoods are either untouched or dropped, but NOT rescaled. We have to argue that any rescaling on the distribution especially the high-likelihood part is very risky. We have also stated in Section 4.2 that the rescaling of our algorithm breaks the monotonicity of the original distribution and directly violated two properties [4] by Nadeem et al. (2020) but counter-intuitively results in betters metrics (closer to the human point in Figure 12b and higher novelty score in Table 1 and Figure 17). This is the crucial point of this work to testify since a shallowly designed manipulation on the high-likelihood part might drastically lower the quality of the text despite a destructively huge increase in diversity. For this point, we only choose the three property-following baselines that are widely acknowledged as golden methods like most existing works [1,3,4,5] to testify our rescaling mechanism.
>
> --Weakness 3 (lack of description). We apologize that due to the limitation of space we have to omit existing analysis from the references. The token level distribution in [1] is rather an inspiration to our method (actually we released our method in Mar. 2021 on https://arxiv.org/abs/2103.07649, maybe around the same time as [1]), and empirical results have proven the effectiveness of our approach. We hope these issues do not hurt the validity of our work.
>
> --Weakness 4 (difficult understanding of human evaluation). Again, we have to apologize that a great number of details are presented in Appendix A which fully explains the human evaluation experiment. We also have to argue that the human evaluation of the sampling algorithm is notoriously difficult to obtain reproducible results and could be much more complicated than it seems to be. We visually explain the quality variation issue in Figure 13, which will easily create an unfair comparison scenario. Consequently, we specifically address this issue in detail in Section 4.3 and Appendix A to guarantee the rigorousness of the experiment.
>
> Again, we sincerely thank the reviewer’s highly professional and insightful comments. Due to the limitation of space, a great number of important details are presented in the Appendices. We hope our replies could clarify your concerns.
>
> [1] Zhang et al., Trading off diversity and quality in natural language generation.
> [2] Meister et al.,  Locally Typical Sampling.
> [3] Basu et al.,  MIROSTAT: A neural text decoding algorithm that directly controls perplexity.
> [4] Nadeem et. al., A systematic characterization of sampling algorithms for open-ended language generation.
> [5] Holtzman et. al., The curious case of neural text degeneration.

---

### Author Response · Authors · 2022-11-16
**General Response**

We sincerely thank all reviewer’s valuable comments. Due to the limitation of space, many details are presented in the Appendices which could have addressed many concerns of the reviewers. We make the following revisions to the draft.

1. We replace results in Table 1 from 300 seconds to 350 seconds as the thresholds of time consumption on the HIT responses. Both results were originally included in Figure 17 in Appendix A. The 300s case creates an impression of insignificant advantage to our methods. We explain this in *Response to Review yD5T* as well as in Appendix A.

2. We replace the faulty use of the word “regularization” with “normalization” which occurs twice in the draft in Section 3.

---

### Decision · Program_Chairs · 2023-01-20

**Decision:**

Reject

**Justification For Why Not Higher Score:**

Already explained in the meta-review.

**Justification For Why Not Lower Score:**

N/A

**Metareview: Summary, Strengths And Weaknesses:**

This paper proposes a sampling strategy to improve the quality and diversity of open-ended generated text, using inverse probability weighting. While this is a relevant problem, there are several weaknesses in this paper as pointed out by all the reviewers: lack of theoretical motivation for the proposed techniques (which look a bit more like heuristics), not enough information about hyperparameter tuning, lack of comparison with other techniques proposed in the literature. The authors revised the manuscript but this was not sufficient to address the main concerns. I therefore recommend rejection, and I hope the authors take into account the reviewers comments in a future iteration of their work.